# Dynamic and Functional Alterations of Neuronal Networks In Vitro upon Physical Damage: A Proof of Concept

**DOI:** 10.3390/mi13122259

**Published:** 2022-12-19

**Authors:** Sàlem Ayasreh, Imanol Jurado, Clara F. López-León, Marc Montalà-Flaquer, Jordi Soriano

**Affiliations:** 1Departament de Física de la Matèria Condensada, Universitat de Barcelona, E-08028 Barcelona, Spain; 2Universitat de Barcelona Institute of Complex Systems (UBICS), E-08028 Barcelona, Spain

**Keywords:** neuronal cultures, functional organization, network damage, resilience, recovery

## Abstract

There is a growing technological interest in combining biological neuronal networks with electronic ones, specifically for biological computation, human–machine interfacing and robotic implants. A major challenge for the development of these technologies is the resilience of the biological networks to physical damage, for instance, when used in harsh environments. To tackle this question, here, we investigated the dynamic and functional alterations of rodent cortical networks grown in vitro that were physically damaged, either by sequentially removing groups of neurons that were central for information flow or by applying an incision that cut the network in half. In both cases, we observed a remarkable capacity of the neuronal cultures to cope with damage, maintaining their activity and even reestablishing lost communication pathways. We also observed—particularly for the cultures cut in half—that a reservoir of healthy neurons surrounding the damaged region could boost resilience by providing stimulation and a communication bridge across disconnected areas. Our results show the remarkable capacity of neuronal cultures to sustain and recover from damage, and may be inspirational for the development of future hybrid biological–electronic systems.

## 1. Introduction

Brain-inspired systems use the functioning principles of biological neurons and synapses, as well as brain architectural traits, to design electronic circuits and software algorithms able to carry out complex human-like tasks, from fluent communication to autonomous decisions. Artificial intelligence, machine-learning and cognitive cybernetics are brain-inspired systems that have an everyday presence in modern society [1]. These systems are also gaining upmost importance in the context of human–machine interfacing and bioelectronic medicine [2,3], e.g., for the development of implants [4] or brain-controlled robotic arms [3].

The development of reliable implants and robotic limbs faces the challenge that biological and electronic circuits have to blend with one another and exchange information reliably [5,6]. Additionally, such biohybrid systems face the inherent dangers of biological damage or electronic materials degradation. Thus, research has focused on accessible lab-on-chip systems to investigate in a controlled manner the capabilities, resilience and limitations of hybrid bioelectronic devices. Pioneering studies have used neuronal cultures, i.e., biological neurons grown on glass or multi-electrode arrays, to interact with robotic arms [7] or simulated game-worlds [8]. In this regard, an aspect that has been poorly explored is whether the performance of these interfacing neuronal cultures is compromised when they experience severe damage, for instance due to disease, environmental toxicity or physical injury.

Thus, to understand the resilience of neuronal cultures to damage, here, we designed two proof-of-concept experiments to evaluate the impact of neuronal loss and connectivity disruption on the overall activity and functional organization of rat cortical neurons grown in vitro. These cultures are well known for their self-organization potential [9,10] and capacity to exhibit spontaneous activity [11,12], and have indeed become some of the most used model systems in a variety of scientific disciplines [13,14,15], most notably medicine, bioengineering and physics of complex systems. Since neuronal cultures lack sensory input and task-oriented behavior, as occurs in the brain, the concept of ‘functional organization’ refers to the set of network topological traits linked to the spontaneous activity of the cultures. These topological traits may vary depending on the details of the culture, e.g., layout of connections [15,16] or biochemical environment [14]. Only when neuronal cultures are coupled to systems that can both act on and receive feedback—such as the aforementioned robotic arms and game-worlds [7,8]—does the concept ‘function’ have a similar meaning to that of the brain.

Neuronal cultures are typically grown forming either an ensemble of interconnected neuronal aggregates [17,18] or a homogeneous layer of neurons [11,19]. In our study, we considered both culture types. On the one hand, aggregated cultures exhibit modular characteristics [18,20] and are convenient to identify and target those aggregates that are central for information flow and network functionality. We could therefore assess, upon removal of specific aggregates, the impact of a targeted attack on the behavior of modular networks. On the other, homogeneous cultures show strongly synchronous dynamics, in which all neurons activate together or remain silent [11]. These cultures were ideal to inspect the impact of a large injury that cut most of the neuronal network in half, thereafter evaluating the effect on synchronization as well as the capacity of the cultures to restore functionality. For both culture types, we observed that the neuronal networks could restore their pre-damage activity levels and preserve their major functional traits. Our results provide an interesting starting point to quantify resilience in neuronal networks, and open new avenues to understand the capacity of biological circuits to maintain operational control, under adverse conditions, on electronic devices that depend on them.

## 2. Materials and Methods

### 2.1. Neuronal Cultures

All procedures were approved by the Ethical Committee for Animal Experimentation of the University of Barcelona (CEEA–UB) under order B-RP-094/15-7125. In all experiments, rat cortical neurons from 19-day-old Sprague–Dawley embryos were used; following the protocols described in Refs. [21,22], dissection was carried out in cold L-15 medium supplemented with glucose (3%) and gentamicin (0.2%) (Sigma-Aldrich, St. Louis, MO, USA). Embryonic cortices were dissociated by repeated pipetting and neurons were plated on substrates that were either poly-dimethyl-siloxane (PDMS) discs or glass coverslips, as detailed later. Glass coverslips were first cleaned in a 70% nitric acid solution for 2 h, rinsed with double distilled water and sonicated in ethanol. Once dried, these substrates were, in turn, placed on 4-well culture plates for ease of access and manipulation. Each well contained plating medium (90% Eagle’s MEM (Minimum Essential Medium)—enriched with 0.6% glucose, 1% 100× glutamax (Gibco, ThermoFisher Scientific, Waltham, MA, USA) and 20μg/mL gentamicin—with 5% horse serum, 5% fetal calf serum and 1μL/mL B27) to ensure the healthy development of the neurons, which were seeded with a density of about 400 neurons/mm${}^{2}$ together with glial cells, and incubated at 37 °C, 95% humidity and 5% CO${}_{2}$. Neuronal density was quantified by capturing high-resolution fluorescence images in different regions of the culture and by taking advantage of the neuronal specificity of the fluorescence probe used, as described later. Additionally, and based on previous studies, there was typically 1 glial cell per every 10 neurons [23]. After four incubation days, medium was switched to changing medium (90% enriched MEM, 9.5% horse serum and 0.5% FUDR (5-fluoro-deoxy-uridine)) for three days to limit glia growth, and then to the final culture medium (90% enriched MEM, 10% horse serum), which was periodically replaced every 3 days [21,22]. Neurons self-organized and reconnected within 48 h to form de novo networks that exhibited spontaneous activity by day in vitro (DIV) 5.

Two types of cultures were prepared, namely, *aggregated* and *homogeneous* (Figure 1a). Aggregated cultures [18] shaped islands of densely packed neurons that could be easily identified and targeted, shaping a network configuration that was ideal for localized damage in modular networks. Homogeneous cultures [19] gave rise to a layer of neurons that uniformly covered the substrate, a configuration that was suited to apply a cut all along the diameter of the culture.

For the *aggregated cultures* (Figure 1a, top), neurons were seeded on bare glass coverslips (#1 Marienfeld-Superior, Lauda-Königshofen, Germany), leading to a strong spontaneous aggregation that shaped compact islands weakly adhered to the glass. In order to visualize the entire network, the available area for neuronal development was restricted to a 6 mm diameter well by using a ring of PDMS 2 mm high previously attached to the glass.

For the *homogeneous cultures* (Figure 1a, bottom), neurons were seeded on PDMS discs 6 mm in diameter that were previously bonded to glass coverslips 13 mm in diameter. To ensure a homogeneous layer of neurons, the combined glass–PDMS systems were previously coated with poly-d-lysine (PDL) [24] and left for 24 h in the incubator, rinsing them with double–distilled water (DDW) prior to neuronal seeding. Different heights of PDMS discs were explored in order to investigate the interaction between the neurons on top of the disc and those on the glass. The latter effectually shaped a population that *surrounded* the disc. The different heights *h* were obtained by pouring preset quantities of PDMS into a plastic Petri dish.

Three kinds of homogeneous cultures were considered depending on the height *h* and other characteristics. A first one with h=0.7 mm shaped a network in which the neurons on the disc connected with the bottom glass, influencing one another. A second configuration with h=2 mm provided very weak PDMS–glass interconnectivity; however, for clarity in the interpretation of results, the neurons on the glass were manually removed (sketch of Figure 1a, bottom) to completely isolate the neuronal network on the disc from its surroundings. A third configuration with h=3 mm ensured a full disconnection between disc and glass neuronal populations.

### 2.2. Fluorescence Calcium Imaging

The genetically encoded fluorescence calcium indicator GCaMP6s [25] (100843-AAV9, Addgene, Watertown, MA, USA) was used to visualize neuronal activity on the same cultures along different days. GCaMP6s contains a vector driven by Synapsin I promoter (Syn I) that is only expressed in mature neurons. The indicator was incorporated into the cells through an adeno-associated virus serotype 9 (AAV9, Titer $\geq 1\times 10^{13}$ vg/mL) at DIV 2.

Calcium imaging was carried out on a wide-field fluorescence microscopy setup (Zeiss Axiover 25C, Zeiss GmbH, Oberkochen, Germany) that incorporated a high-speed camera (Hamamatsu Orca Flash 2.8, Hamamatsu Photonics, Hamamatsu city, Japan) together with a light source for fluorescence (mercury vapor arc lamp, Osram GmbH, Munich, Germany). Series of fluorescence images were recorded at 50 frames/s, 8-bit gray-scale levels and a size of 1024×1024 pixels. In combination with a 2.5× objective and an optical zoom, an area of 7.1×7.1 mm${}^{2}$ could be accessed, sufficient to monitor a 6 mm diameter culture and its surroundings. Spontaneous activity recordings were typically 15–30 min long and were acquired using the Hokawo 2.5 software (Hamamatsu Photonics, Hamamatsu, Japan).

Cultures were recorded inside a glass micro incubator (Ibidi, GmbH, Gräfelfing, Germany) that maintained the same environmental conditions as in the standard incubator except for the temperature, which was set to 25 °C to favor a higher spontaneous activity.

### 2.3. Damage Schemes

The strategies to induce damage in the neuronal cultures were different for the two culture types, as described below.

*Damage in aggregated cultures.* Damage was carried out by sequentially removing a given neuronal aggregate of the approximately 100 aggregates present in the network, and according to network topological criteria described later. There were typically 8 chained damaging actions on the culture, one per day, to allow the culture to partially recover in between actions. Damage actions extended from DIVs 9 (onset of strong spontaneous activity) to 19 (beginning of culture degradation or lack of activity). The criterion to select the aggregates to be damaged were established to compare *targeted attack* and *failure* [26]. For targeted attack, the selected aggregates were those with important functional roles, either because they funneled information flow (high centrality) or because they were topologically important (highest degree). For failure, the damage aggregates were simply taken randomly. Thus, three experimental schemes were delineated: centrality attack, degree attack and random deletion.

For each scheme, spontaneous activity in a culture was recorded for 20 min before the application of any damage, and data swiftly analyzed to determine the aggregate of interest, which was then removed or disconnected from the network by using a needle (Figure 1b). The culture was next recorded again for another 20 min and analyzed to investigate the degree of inflicted damage. The culture was thereafter allowed to recover for 24 h and the damage process repeated again.

*Damage in homogeneous cultures.* Damage consisted of cutting diametrically the neuronal population on the 6 mm PDMS disc with a scalpel at DIV 15 to later follow the recovery of the network in quasi-logarithmic time intervals. The inflicted wound was typically 500 μm wide and was consistent among repetitions. DIV 15 was selected as the initial day since neurons had already formed a well-interconnected network, with neurons stable in fixed positions and homogeneously spread over the surface. A total of six cultures were considered, two for each preset height *h*. For each pair, a culture was damaged and the other was left as control. Before applying any damage, spontaneous activity was recorded for 15 min. The cut was next made on the culture, effectually separating the network on the PDMS disc in two parts, and activity recorded again for 30 min (Figure 1c). Then, cultures were recorded over 15 min in the following preset times: 2 h, 6 h, 24 h and 3 days.

### 2.4. Image Analysis

We used NETCAL [27], a Matlab-built toolbox, to analyze calcium fluorescence recordings. Bright objects on the images were selected as regions of interest (ROIs), and the raw fluorescence trace of each object *i* as a function of time, $F_{i}\left(t\right)$, was extracted. In aggregated cultures, there were approximately 100 ROIs in each experiment, each ROI corresponding to an aggregate. For homogeneous cultures, the field of view was discretized onto a 40×40 square grid of ROIs, totaling 1300 ROIs that approximately contained 3 to 5 neurons.

Regardless of the culture type, the obtained fluorescence traces were normalized as $\Delta FF_{i}\left(t,\%\right)\equiv 100\cdot \left(F_{i}\left(t\right)-F_{0,i}\right)/F_{0,i}$ for each ROI *i*, where $F_{0,i}$ is the background fluorescence (neurons at rest). Finally, fluorescence data were transformed into a binary signal using the Schmitt trigger method [28], which identifies an activity event whenever sharp increases in the fluorescence signal occur. The resulting time series contained a ‘1’ when activity was present at time *t* and ‘0’ otherwise. Network activity was visualized in the form of raster plots (Figure 2).

### 2.5. Activity Analysis

Binary time series were analyzed in Matlab to extract a number of descriptors related to the overall dynamics of the network. This included the identification of *network bursts*, i.e., activity events in which a substantial group of ROIs activated together in a short time window. Such a bursting behavior was strong in homogeneous cultures and practically encompassed the entire network [19]. For aggregated cultures, bursts were identified as groups of aggregates that activated in a cascading manner with few ms from one to another [18] and typically encompassed 10–20% of the network. In either case, the time spanned between consecutive bursts was denoted *inter-burst interval* (IBI) and was a convenient indicator to reveal dynamic alterations due to damage. Since neuronal bursts can be viewed as the firing frequency of the culture, *network activity A* was simply computed as the inverse of the average IBI.

### 2.6. Transfer Entropy and Effective Connectivity

Causal interactions between pairs of ROIs were inferred using Transfer Entropy [29,30] and computed in Matlab by means of an implementation suited for calcium imaging recordings known as Generalized Transfer Entropy [31]. Using the trains of binarized activity data, an effective connection from ROI *I* to *J* (${\mathrm{TE}}_{I\to J}$) was established whenever the information contained in *I* significantly increased the capacity to predict future states of *J*. Instant feedback was present, and Markov Order was set to 2 [31]. The significance threshold *z* for effective connections was established by comparing the transfer entropy estimate ${\mathrm{TE}}_{I\to J}$ with the joint distribution of all input *X* to *J* and output *I* to *Y* (for any *X* and *Y*) as
(1)$$z=\frac{{\mathrm{TE}}_{I\to J}-\langle {\mathrm{TE}}_{\mathrm{joint}}\rangle }{\sigma_{\mathrm{joint}}},$$
where $\langle {\mathrm{TE}}_{\mathrm{joint}}\rangle$ is the average value of the joint distribution and $\sigma_{\mathrm{joint}}$ is its standard deviation. Significant connections were then accepted for z≥1. This threshold allowed to capture effective communication both at global and local scales with a sufficient number of connections present for reliable analysis. Accepted connections were finally binarized as 0 (absence of connection) or 1 (connection present), shaping directed yet unweighted connectivity matrices. These matrices were visualized in the form of network maps with Gephi [32]. For clarity of language, here, we use the term ‘effective’ when referring to connections and ‘functional’ when referring to overall network organization.

### 2.7. Centrality and Network Properties

Effective connectivity matrices were analyzed in the context of complex networks [33] to evaluate the impact of damage in the overall network functional organization. The following network properties were evaluated.

*Degree k.* Accounted for the number of connections of every ROI *i* as $k_{i}=\sum_{j}A_{ij}$, where $A_{ij}$ is the adjacency matrix of the network. Since the functional networks were directed, the total degree contained both the incoming $k_{i}^{\mathrm{in}}=\sum_{j}A_{ji}$ and outgoing $k_{i}^{\mathrm{out}}=\sum_{j}A_{ij}$ links; therefore, $k_{i}=k_{i}^{\mathrm{in}}+k_{i}^{\mathrm{out}}$.

Those ROIs in the network with the highest degree were important since they either received information from or transmitted information to the rest of the network; thus, they were considered *central*.

*Global Efficiency $G_{\mathrm{eff}}$.* Captured the capacity of the network to exchange information as a whole [33] and was given by
(2)$$G_{\mathrm{eff}}=\frac{1}{N(N-1)}\sum_{i\neq j}^{N}\frac{1}{d_{ij}},$$
where *N* is the total number of ROIs and $d_{ij}$ the shortest path length, i.e., the minimum number of effective connections that communicate any two nodes *i* and *j*,
(3)$$d_{ij}=\sum_{L_{uv}\in g_{i\to j}}L_{uv},$$
with $L_{uv}$ representing the geodesic path $g_{ij}$ between every pair of nodes, computed using the Dijkstra algorithm. The computation of $G_{\mathrm{eff}}$ was carried out in Matlab using the Brain Connectivity Toolbox (BCT) [34]. $G_{\mathrm{eff}}$ varied between 0 (an ensemble of isolated neurons) and 1 (all-to-all effective connectivity, swift information flow).

*Modularity Q.* Quantified the tendency of ROIs to form functional communities, i.e., ensembles of ROIs more connected within their group than with the rest of the network, and was defined as
(4)$$Q=\frac{1}{2m}\sum_{ij}\left(A_{ij}-\frac{k_{i}k_{j}}{2m}\delta \left(c_{i},c_{j}\right)\right),$$
where $A_{ij}$ is the adjacency matrix of the network; $k_{i}$ and $k_{j}$ are the sum of connections bound to ROIs *i* and *j*, respectively; *m* is the total number of connections; δ is the Kronecker delta function; and $c_{i}$ and $c_{j}$ are the communities that nodes *i* and *j* belong to, respectively. The modularity was calculated using the Louvain algorithm [35] provided by the BCT. *Q* varied between 0 (the entire network is the sole community) and 1 (isolated ROIs). Typically, Q≳0.3 indicated the presence of modular organization.

*Betweenness centrality BC.* Accounted for the importance of a particular ROI in the flow of information across the network. It was determined as
(5)$${\mathrm{BC}}_{i}=\sum_{j\neq k}^{N}\frac{n_{jk}\left(i\right)}{n_{jk}},$$
where $n_{jk}$ is the number of shortest paths $d_{jk}$ that link *j* to *k*, and $n_{jk}\left(i\right)$ the number of shortest paths connecting *j* and *k* that pass through *i*. BC was an important centrality measure since it identified those ROIs that routed most of the network communication.

## 3. Results I: Damage on Aggregated Cultures

We investigated the impact of targeted attack and failure in cultures that contained about 100 aggregates of densely packed neurons (Figure 1a). Targeted attack actions consisted in sequentially eliminating nodes in the network that were *functionally central*, either because they had the highest degree *k* or the highest betweenness centrality BC (Figure 1b). Central aggregates were determined by analyzing the effective connectivity of spontaneous activity recordings and by inspecting thereafter the network properties of the culture under study. The most important central aggregate was then removed from the culture with a needle and spontaneous activity recorded again to observe the effect of damage. The entire process was repeated again 24 h later. A complete sequence of damage typically encompassed 7–8 steps of aggregate deletion, and was stopped when the spontaneous activity became too weak or the cultures too old for reliable analysis. Targeted attack actions were compared with random deletions of aggregates (*failure*) to pinpoint the importance of centrality in network vulnerability.

We explored a total of nine aggregated cultures, with three repetitions for each damaging action (two target attacks and one failure). A first and rather surprising observation was that the level of spontaneous activity abruptly dropped for the first minute just after damage, then boosted up for 15–30 min and finally relaxed towards pre-damage levels within a few hours. This boost in activity is illustrated in Figure 2a; it was ascribed to the activation of homeostatic response mechanisms in the network [22] that attempt to stabilize activity.

The ratio in spontaneous activity after damage and before it, $\Phi =A_{\mathrm{after}}/A_{\mathrm{before}}$, is shown in the form of box plots in Figure 3a for the three damage actions. From the plots, first, we note that dispersion in data was strong and that the ratio Φ could vary from Φ≃0.4 (decrease in activity) to Φ≃2 (increase in activity by almost 2-fold); second, and more importantly, the increase in Φ was significantly more pronounced in failure than in targeted attack on betweenness centrality (BC) or degree *k*, with the latter two exhibiting similar behavior. These results indicate that targeted attack on aggregates that are pivotal for information flow hinders the capacity of the network to swiftly respond to damage. Additionally, since the results in both targeted attacks (BC or degree) are similar, we take the attack on BC as the representative scenario for damage on central nodes.

We next studied the functional organization of the explored networks through Transfer Entropy and computed the global efficiency $G_{\mathrm{eff}}$. This network property captures the capacity for swift information traffic across the network; therefore, it provides insight on global network affectations due to damage, particularly in targeted attack. Since $G_{\mathrm{eff}}$ depends on the number of nodes *N* in the network, we excluded the deleted node in all computations of $G_{\mathrm{eff}}$ so that the functional networks before and after aggregate deletion had the same size.

As shown in Figure 3b for attack on BC, the capacity of the network to exchange information is reduced after each damage step (cyan bars are shorter than maroon ones). The decrease in $G_{\mathrm{eff}}$ is small, by about 10%, but consistent along the damage sequence. This is important as it indicates that targeted attack on BC compromises the capacity of the network for global communication. We note, however, that the time span between consecutive damage steps is 24 h and that this period was sufficient for full recovery of global network communication, which even exceeded previous values. Indeed, the maroon bars of Figure 3b, left, are progressively higher in damage steps 1 to 4. Only when damage is very extended, typically after seven damaging steps, is the drop in $G_{\mathrm{eff}}$ severe.

By contrast, the evolution of $G_{\mathrm{eff}}$ for random attack (Figure 3b, right) shows a strong variability along the sequence, with changes between pre- and post-damage that are much more abrupt as compared to targeted attack. For instance, in the first damage step, $G_{\mathrm{eff}}$ almost doubles just after node deletion, and practically triples upon recovery. We also observed that, when inspecting the behavior of the cultures in different experimental repetitions, the first step in the damaging sequence for random attack always led to a boost in $G_{\mathrm{eff}}$ (cyan bar above maroon one), a trait that was not observed in targeted attack either on BC or degree.

A complementary approach to explore the impact of damage on network functionality consisted in analyzing the alterations in modular organization. For that, we computed the modularity *Q* before and after damage, and compared again the three possible actions, i.e., attack on BC, attack on degree and failure. As shown in Figure 4a, both targeted attacks led to an average increase in modularity of 10%, while failure led to a reduction in modularity of 5%, with statistical significant differences between attack on BC and failure. The increase in modularity after damage in the targeted attack scenarios strengthens the message that network global communication was affected, with aggregates tending to communicate at more local scales. This is illustrated in the toy network of Figure 4b, in which the deletion of the top BC node increases *Q* by about 25%. This important change is associated to the role of the node, which connects different communities through its shortest paths, causing a higher isolation of the communities when removed.

A representative example of the alterations caused in the neuronal cultures upon targeted attack on BC is shown in the network maps of Figure 4c. Before damage, we can observe a dense mesh of effective connections that render a network with high $G_{\mathrm{eff}}$ and moderate *Q*. The density of connections decays after the deletion of the aggregate with the highest BC (white outline), leading to communities that are more isolated, particularly the yellow one. These changes are reflected in an increase in *Q* of 12% and a decrease in $G_{\mathrm{eff}}$ of 13%. The corresponding adjacency matrices of the network maps are shown in Figure 4d. The black dots are effective connections and the colored boxes represent functional communities. The effect of a loss in global network integration is captured here by a substantial decrease in the number of connections outside the color boxes. Although the modular organization is maintained, with the number and size of communities similar in the two matrices, the isolation of the modules themselves is noticeable, overall shaping a more segregated network.

## 4. Results II: Damage on Homogeneous Cultures

Here, we considered neurons that grew covering in a uniform manner the surface of a 6 mm diameter PDMS disc, which, in turn, was attached to a 13 mm diameter glass coverslip (Figure 1a). The damage action consisted in cutting diametrically the neuronal population on the disc in two halves with a scalpel and investigating its subsequent recovery. The height of the disc could be adjusted to control the connectivity between the neurons on the disc and those on the glass. Heights of about h≃0.7 mm or smaller ensured a connectivity between top (disc) and bottom (glass) neuronal populations, with the latter effectually shaping a reservoir of healthy neurons that was never perturbed during the damage action. One of our interests lay precisely on understanding whether this *surrounding* of neurons played a role in network recovery. Heights of about h≃3 mm or higher ensured a disconnection between top and bottom populations. This configuration could be alternatively prepared by gently removing the neurons on the glass that were in contact with the disc a few days after plating, giving rise to a population of neurons in the surroundings that acted independently (Figure 1a, dashed rectangle).

Spontaneous activity in the glass–PDMS system was recorded before damage, immediately after it and in the subsequent times that followed a quasi-logarithmic scale. To facilitate the analysis, the entire field of view was divided onto a 40×40 square grid of ROIs, providing 1300 ROIs in total. As shown in Figure 1c for a design with h=0.7 mm, neurons before damage spontaneously activated in a bursting manner, with the entire population lighting up synchronously. The action of damage left a culture split in two, with each half ‘A’ and ‘B’ (about 30% of ROIs each) activating spontaneously on its own. We observed that the surroundings ‘S’ (neurons on the glass, about 25% of ROIs) were also substantially active. Additionally, we noticed that a very small population of neurons grew in the walls of the PDMS disc, shaping a contour ‘C’ (5% of ROIs) that could not be accessed by the scalped and that maintained activity after damage.

The overall impact of damage on network collective dynamics is illustrated in the raster plots of Figure 2b. The neat bursting behavior before damage is visually captured by vertical bands that extended all ROIs. After damage, however, dynamics splits into a varied mixture of patterns in which synchronous bursting only occurs within each population.

To investigate the recovery of the network, we computed the average activity *A* in each population (regions A, B, C and S), together with a control experiment—recorded in identical conditions but left undamaged. Since synchronous bursting was maintained in each of these populations and the control experiment, the activity *A* was simply determined as the number of bursts present along a spontaneous activity recording (typically 15 min).

Figure 5 provides the results of the analysis of activity for three representative experiments, each one with a different height *h* and with the surroundings ‘S’ present or isolated. For h=0.7 mm and surroundings present, activity in the different regions remains similar to the control, with the exception of a half region of the disc (region ‘A’) that boosts in activity. This similarity with the control case is in general maintained along the next hours and days, and suggests that the surrounding of unaltered neurons may play the role of a pacemaker that maintains the entire system’s stability in terms of dynamics. On the contrary, for h=3 mm and weak coupling with the surroundings, the different populations show a weaker activity and, in general, depart from control, although the variability in activity across populations within the damaged culture seems to reduce at late stages, possibly indicating a restoration in the connectivity between regions. For the situation with h=2 mm and surroundings manually isolated, the regions also show weaker dynamics as compared to controls, as well as a strong variability across regions. We note, however, that in the three experiments, the region that departs more strongly from the rest within the same culture is the contour ‘C’, which is just 5% of the ROIs monitored. Thus, it is enlightening the observation that the majority of monitored ROIs tend to converge toward similar levels of spontaneous activity and that very rarely do any of the monitored regions become silent. This maintenance of activity indicates that homogeneous cultures can cope well with damage and homeostatically adjust themselves to preserve the frequency of spontaneous activations.

To complete the study, we inspected the functional organization of the three representative experiments. Functional networks were extracted using Transfer Entropy; however, since activity was always synchronous within a region of the culture, we found it convenient to inspect functional interactions just between the four main regions (halves ‘A’ and ‘B’, contour ‘C’ and surroundings ‘S’) rather than between all the 1300 ROIs.

The results for functional connectivity are provided in Figure 6. The panels show, for each of the three damage experiments, the interaction of the different regions before damage (green), just after damage (red outline) and in subsequent time steps (alternating pink and white). The panels also compare the damaged network with the corresponding control. For each experiment, the regions are shown as colored circles whose areas are proportional to their number of ROIs. The blue bands connecting the regions are proportional to the number of effective connections that the regions share. Thus, the thicker the band, the stronger the exchange of information between regions. Overall, the most important interactions occurred between the two halves ‘A’ and ‘B’ as well as with the surroundings ‘S’. The role of the contour ‘C’ was negligible.

For the cultures with h=0.7 mm and strong coupling with the surroundings, we can observe that there is a strong communication between the two halves ‘A’ and ’B’ of the culture as well as with the surroundings ‘S’. Damage disrupted most of the communication between ‘A’ and ‘B’, but the interaction with ‘S’ was maintained, and even preserved during recovery. The two halves ‘A’ and ‘B’ were disconnected after damage but functionally reconnected after 6 h. Given the strong involvement of the surroundings ‘S’ in preserving overall functionality, we hypothesize that they mediated in recovery of communication between the two halves of neuronal populations on the disc. The control experiment, as expected, maintained an overall strong connectivity all along the exploration.

The above hypothesis was verified by looking at the experiment with h=3 mm, in which halves ‘A’ and ‘B’ are strongly coupled before damage but totally disconnected from ‘S’. Damage separated the two halves, and no substantial reconnection occurred throughout the rest of the recording times. The control experiment for this case maintained a strong coupling between ‘A’ and ‘B’. Finally, for h=2 mm, we observed a similar behavior as for h=0.7 mm, indicating that the surroundings were not as neatly isolated as expected. This experiment, however, is interesting, since it indicates that even a weak presence of a reservoir of neurons around the focus of damage facilitates functional reconnection.

## 5. Discussion

Here, we presented two main experimental approaches to deliver damage in neuronal cultures and study their subsequent recovery. The two considered cultures, namely, aggregated and homogeneous, were different in neuronal spatial organization and dynamical traits but both exhibited a remarkable capacity to maintain activity after injury and to preserve their large-scale functional characteristics. Indeed, the aggregated networks retained high levels of global efficiency $G_{\mathrm{eff}}$, a property linked to whole-network communication capacity, after eight targeted node deletions, while the homogeneous networks kept their high levels of synchronization within the separated regions after the cut, communicating with one another through the reservoir of untouched cells located at the surroundings of the PDMS disc. We observed that the damage experiments were in general reproducible as far as the damage protocol was maintained. In this sense, recovery may be faster or more extended when young cultures are used, e.g., by DIV 5, when they are in a strong process of axonal growth and synaptic formation. Additionally, and for the particular case of homogeneous cultures, recovery could substantially change when exploring wounds of different widths. The scalpel-induced cut separated the two halves of the culture by 500 μm, which could be substantially reduced by using, for instance, a laser [20].

Our experiments were designed as a proof of concept, in the sense that we aimed at demonstrating that response to damage in living neuronal networks could be investigated in detail using tools from complex systems. However, two important aspects, among others, were not covered here and would require attention in future investigations. The first aspect is characterization of the conditions that make spontaneous activity completely cease after repeated damage. Our cultures were always active following damage, but we did not quantify under which conditions they became totally silent and unable to recover. We observed in previous studies that isolated neurons and aggregates were able to show activity by themselves when chemically disconnected [18,19]; therefore, it may not be possible to fully silence the cultures. Network-wide correlated activity may vanish, but the culture may still be operative at a neuron level. The second aspect is the role of glial cells, which are present in our cultures and that, in general, are pivotal for the development of synaptic connections and the maintenance of activity in neuronal circuits [36,37]. Recent studies have shown that glial cells, particularly astrocytes, change their morphology and protein expression upon damage [38,39], which, together with the expression of neurotrophic factors by the neurons themselves upon trauma [40], shape complex molecular pathways at a microscopic level that molds recovery. To frame a complete understanding of damage, one would need to link these microscopic changes to the mesoscopic, network-scale ones.

We chose aggregated and homogeneous cultures as main experimental models because they portray two contrasting yet highly investigated dynamic scenarios. On the one hand, aggregated cultures show a modular behavior both in dynamics and functional organization [18], in the sense that groups of aggregates tend to activate in small groups and form functional communities, a trait that is intrinsic of the brain [41,42] and that is considered a fundamental asset to mimic brain-like behavior in vitro. On the other hand, homogeneous cultures represent the classical, widely studied culture model that is often used in combination with electronic devices. The advantage of homogeneous cultures is that they exhibit a robust, periodic bursting behavior that is difficult to suppress [11,26], therefore providing continuous background activity despite severe physical damage. Interestingly, the observation that a reservoir of healthy neurons surrounding an injured area facilitates recovery suggests that a continuous bombardment of neuronal activity onto a damaged area is indeed important and relates to medical experiments in which rehabilitation from stroke is accelerated when the affected area receives stimulation [43], since it promotes the activation of plasticity mechanisms that restore connectivity.

Related to the above, the modular characteristics of aggregates cultures could provide a general framework for lab-on-chip investigations that model brain damage, e.g., in the context of stroke [44] or neurological disorders [45]. For the latter, it has been argued that central nodes or *hubs* [46] are prominently affected in brain diseases such as Alzheimer’s [47,48,49], and that often several nodes are lost until cognitive or motor deficits emerge. In our experiments, no apparent changes in global efficiency were noticeable until eight aggregates—about 10% of the nodes—were deleted. Our study, in conjunction with related experimental [22,50] and numerical investigations [51,52,53,54], could help in understanding the mechanisms by which the network is capable of such a resilience and find signatures that anticipate network-wide collapse. The challenge is difficult since we must note that the effective network extracted from the recordings may be very different from the underlying structural network. Thus, future experiments should extract information of both effective and structural connectivity representations and explore how they relate to one another.

Finally, in the context of hybrid bioelectronic devices, we hypothesize that an ideal biological design to ensure resilience would be one in which a large homogeneous network was coupled with an aggregated one that provided a modular backbone. The former would ensure periodic background activity, while the latter would approach brain-like organization. The two together could not only ensure the functioning of the hybrid device in case of physical damage, but also help, thanks to modularity, in the brain-like learning and plasticity processes required for swift dialogue between biological and electronic entities. We also note that a fundamental trait of biological circuits is the activation of molecular processes and biochemical pathways in response to damage, making them self-sufficient. This regeneration capacity is absent in electronic circuits; therefore, they have to rely on redundancy strategies and duplication of tasks when they operate in extreme conditions [55], which hinders the optimization of resources and power consumption.

## Figures and Tables

**Figure 1 micromachines-13-02259-f001:**
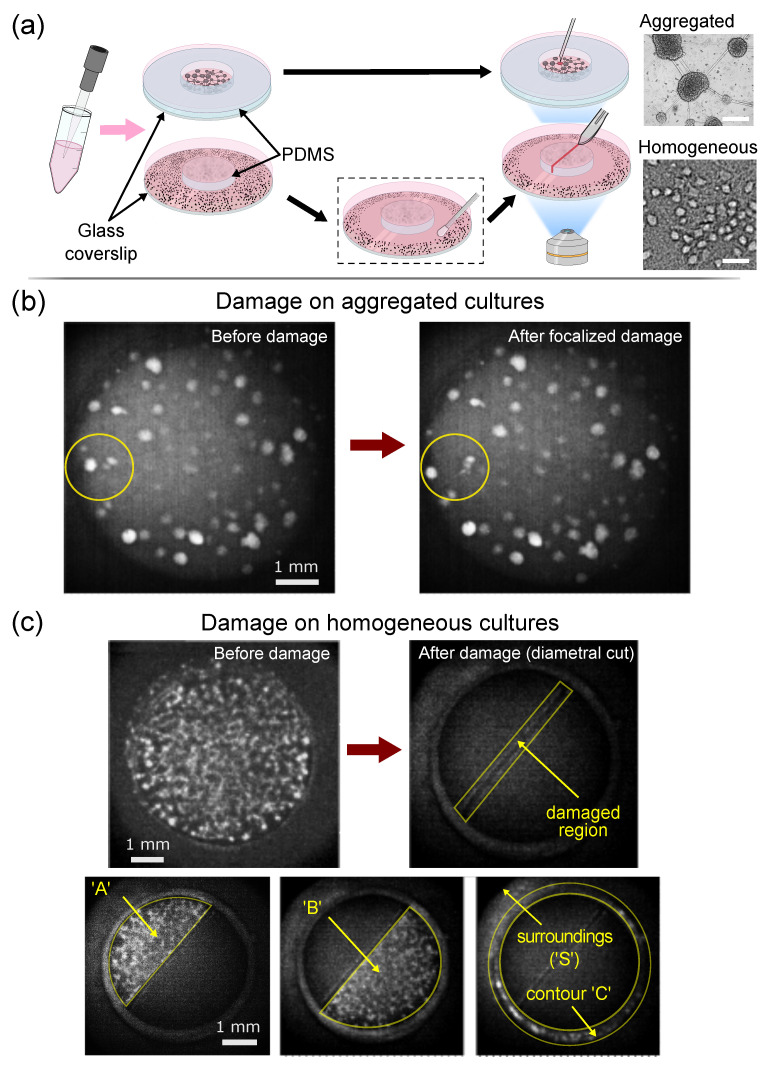
Neuronal cultures and experimental procedure. (**a**) Sketch of the main steps for culture preparation and damage. Neurons grown on bare glass procured ‘aggregated cultures’, in which neurons grouped in compact assemblies (bright field image on the right). These cultures were damaged using a needle. Neurons grown on PLL-coated PDMS discs provided ‘homogeneous cultures’, whose individual neurons could be discerned (bright field image). These cultures were damaged by cutting them in half along their diameter with the help of a scalpel, shaping a wound 6 mm long and 500 μm wide. The PDMS disc was surrounded by a population of neurons that grew on glass and that could be removed if desired (black dash contour). Scale bars are 200 μm. (**b**) Representative fluorescence images of an aggregated culture before and after damage. The yellow outline highlights the damaged region. (**c**) Representative fluorescence images for homogeneous cultures. The top snapshots show the network-wide activation of the culture before damage on the left, and the damaged region upon scalpel cut on the right. The bottom snapshots show the two halves of active neurons that appear 24 h after damage, denoted ‘A’ and ‘B’, as well as the surroundings ‘S’ and contour ‘C’ of the culture.

**Figure 2 micromachines-13-02259-f002:**
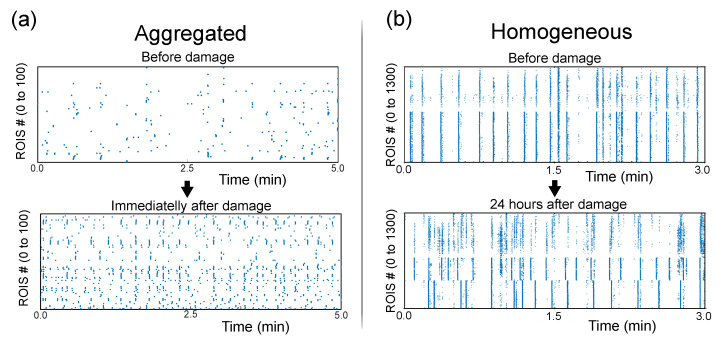
Changes in spontaneous activity upon damage. (**a**) Raster plots of spontaneous activity for an aggregated culture, illustrating the increase in activity immediately after damage. Each blue dot corresponds to the activation of an aggregate. (**b**) Raster plots for homogeneous cultures 24 h after damage, illustrating the rupture of network-wide synchronous activity onto a segregated dynamics within the two halves of the culture. Each dot corresponds to a region of interest (ROI).

**Figure 3 micromachines-13-02259-f003:**
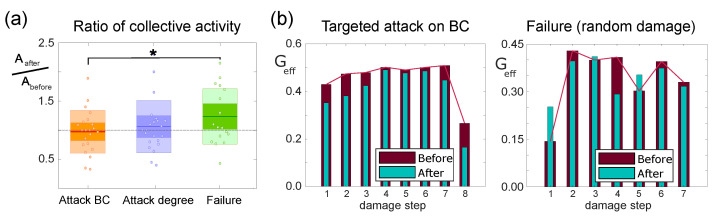
Damage on aggregated cultures. (**a**) Box plot of the collective activity ratio between after ($A_{\mathrm{after}}$) and before ($A_{\mathrm{before}}$) damage for the three different kinds of attack. The dots in each box plot include three experimental repetitions and different damage steps on the same culture. Statistically significant differences are only observed between targeted attack on BC and failure (* *p* = 0.0191, Student’s *t*-test). (**b**) Bar plots representing the evolution of the global efficiency $G_{\mathrm{eff}}$ along a full sequence of damage for a pair of representative experiments, and considering targeted attack on nodes with the highest betweenness centrality (left) and failure (right). For each damage step, the pairs of bars compare the $G_{\mathrm{eff}}$ values before and after the deletion of an aggregate.

**Figure 4 micromachines-13-02259-f004:**
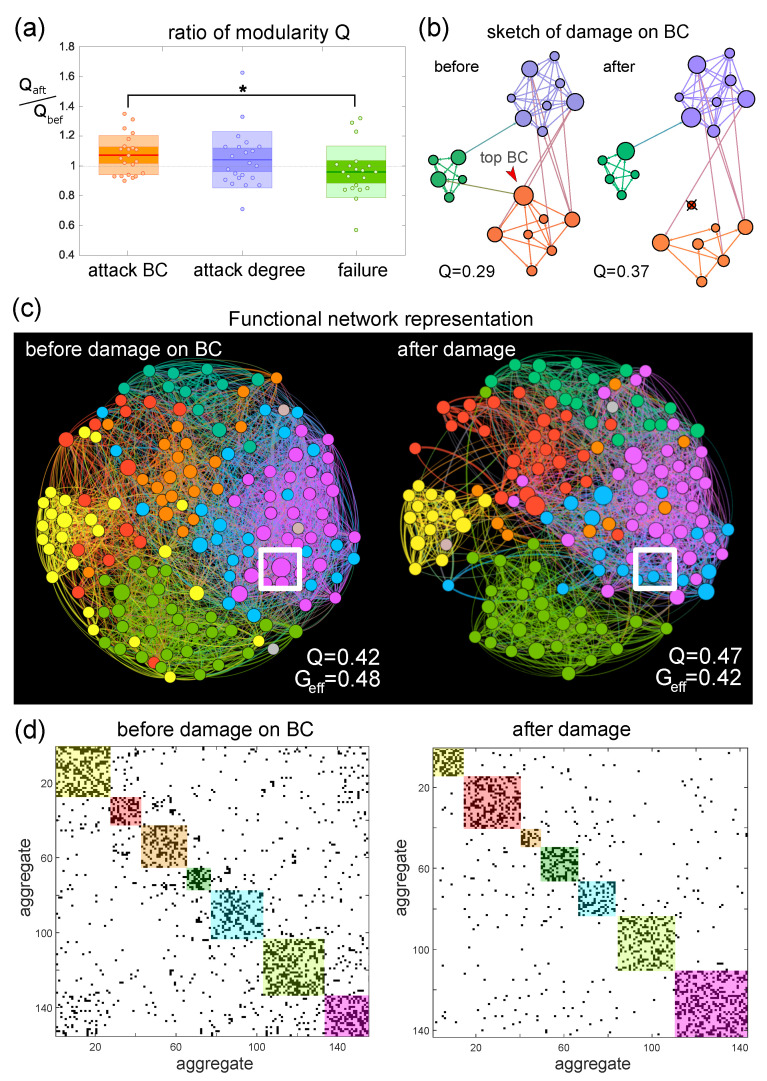
Alterations in modularity upon damage on aggregated cultures. (**a**) Box plot of the modularity ratio between after ($Q_{\mathrm{after}}$) and before ($Q_{\mathrm{before}}$) damage for the three different kinds of attack. Statistically significant differences are only observed between targeted attack on betweenness centrality BC and failure (* *p* = 0.0321, Student’s *t*-test). (**b**) Schematic representation of targeted attack on BC. In the sketch, nodes are colored according to the functional community they belong to, and the diameter of the nodes is proportional to their BC score. The top-scoring node is marked with an arrow and, upon deletion, the network becomes fragmented and the modularity *Q* grows. (**c**) Functional connectivity maps of a representative experiment before and after a targeted attack on BC. The deleted node is marked by a white square. The network becomes overall more segregated, with an increase in *Q* and a decrease in $G_{\mathrm{eff}}$. The nodes in the maps are color-coded according to the functional community they belong to. (**d**) Corresponding connectivity matrices before and after attack. Black dots are effective connections and color boxes are functional communities. The increase in network fragmentation is depicted here by a reduction in the density of effective connections between communities.

**Figure 5 micromachines-13-02259-f005:**
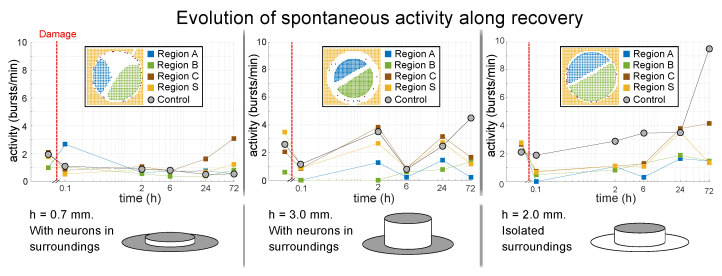
Activity alterations upon damage in homogeneous cultures. The top plots show the evolution of average spontaneous activity (in a quasi-logarithmic scale) for three representative scenarios with varying height *h* of the PDMS disc. The vertical red dashed line marks the time of damage. The bottom sketches provide a visual aid of the experimental preparations, highlighting the presence (gray) or absence (white) of surrounding neurons. The activity is plotted individually for each spatial region monitored (half cultures ‘A’ and ‘B’, contour ‘C’ and surroundings ‘S’). Measurements in a control culture are included as reference.

**Figure 6 micromachines-13-02259-f006:**
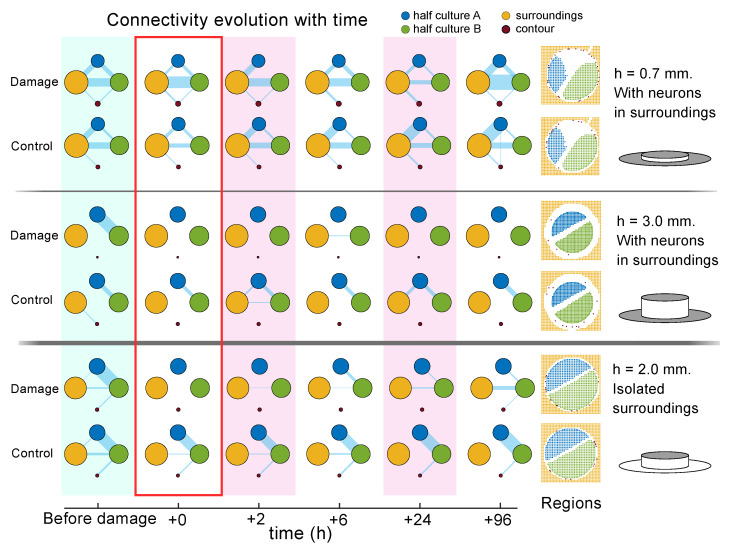
Evolution of functional connectivity upon damage and recovery in homogeneous cultures. The sequence of network diagrams provides three representative damage experiments with varying *h* and connectivity with the surroundings, together with their corresponding controls. The sketches on the right provide a visual aid to the experimental conditions used. Each group of colored circles that shape a network represents the functional organization between the four regions of the culture (halves ‘A’ and ‘B’, surroundings ‘S’ and contour ‘C’) at different stages of the damage sequence. The condition before damage is marked in green, just after damage is marked with a red contour and recovery at different time intervals is marked with alternating pink and white. The area of the circles is proportional to the number of ROIs it contains. The blue bands capture the interaction between regions, with the thickness of the band proportional to the number of shared functional connections.

## Data Availability

Data are available upon request from the authors.

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
