# Peer review of "Dynamic and Functional Alterations of Neuronal Networks In Vitro upon Physical Damage: A Proof of Concept"

_micromachines, 2022, doi:10.3390/mi13122259_

Round 1

Reviewer 1 Report

The study chose aggregated and homogeneous neuronal cultures as the experimental models to investigate the recovery after damage. The results are interesting, but it cannot be accepted in the current form. In addition, there are several aspects that are concerning.

1.     In Materials and methods, the authors seeded with neurons together with glial cells and then prepared the culture in DIV 5. The authors should provide the type of culture medium and the picture of culture. Whether there are neurons and glial cells or neurons only? The repair mechanism after damage may be distinct.

2.     The authors should provide fluorescence calcium images for aggregated and homogeneous neuronal cultures before and after injury.

3.     The authors should discuss the difference in molecular repair mechanism after damage between aggregated, brain-like, and homogeneous, electronic devices-like, neuronal cultures.

4.     Whether the selection of cultures at different time (DIV 5, DIV 10, DIV 20) will affect the experimental results and conclusions?

5.     How to make sure the consistency of damage in different experiments?

Reviewer 2 Report

The paper is very interesting, well-written and the figures are clear and informative. I only have some minor concerns, as follows.

·      It should be acknowledged and addressed in the discussion that basic informational properties of the culture have been assessed and that the ability of the culture to act like a functional neuronal network, i.e. carry out tasks, has not been assessed. Similarly, in both introduction and discussion it would be useful to relate the four network parameters measured to functional neural networks and their ability to carry out tasks, by referencing studies in which these parameters are altered in artificial and biological neural networks and the consequences measured.

·      On a related note, the ability of the two different culture types to preserve their activity could be given more significance by showing what a failure to preserve activity would look like. This is partly addressed by comparing the preservation of global efficiency in one case and inter-region synchronization in the other, but it could be more clearly explored. Importantly, it is clearly not the case that cultured neurons or other biological neural networks can universally preserve and/or recover their functional connectivity and ability to handle information upon being damaged, and thus the lack of a negative control calls into question whether the parameters being measured are really indicative of the functionality of the network.

  • The authors state "Each well contained the proper culture medium to ensure the healthy development of the neurons", but they should specify what media (plus whatever supplements they used), instead of just citing another paper.
  • They say that cells were seeded at a "density of about 400 neurons/mm2 together with glial cells". While it's standard to report how many cells you seed (assuming other researchers reading the paper know how to count cells), they don't clarify how they distinguished neurons from glia in the seeding process to determine how many neurons are plated.
  • I wonder if it is possible to improve the quality of the image in 1C showing damage to the homogeneous culture (I've had to zoom in 300% to see anything at all), maybe by enlarging it or changing the colour/brightness/contrast, etc. Please also report the diameter of the wound made to the homogenous culture (while it's 6 mm in length, the diameter would tell you about the "scale" of the damage, and how separated the two halves of the culture are). This is easily discerned from a micrograph after the scratch has been made.

Minor language issues:

There are a number of minor language issues; the following is not an exhaustive list so I would recommend enlisting a fluent English writer or editing service.

·      Abstract line 6: physically damaged

·      Introduction line 21: should be utmost importance, but I would suggest this phrasing is inappropriate anyway and would just remove or replace with “considerable” or equivalent.

·      Introduction line 25: blend with one another

·      Introduction line 27: research has focused on

·      Introduction line 28: investigated in a controlled manner

·      Introduction line 46: upon removal of specific aggregates

·      Introduction lines 47-48: dynamics is plural so remove “a”

·      Introduction line 50: effect instead of affectation

·      Introduction line 52: pre-damage activity levels

·      Introduction lines 39-40: “the most attractive model systems in a variety of scientific disciplines” is not really a useful statement, as models cannot be meaningfully rated on their “attractiveness” across entire disciplines as broad as medicine, bioengineering and physics of complex systems.

Round 2

Reviewer 1 Report

The paper can be accepted to publish.